# AUTOENCODER IMAGE INTERPOLATION BY SHAPING THE LATENT SPACE

## ABSTRACT

Autoencoders represent an effective approach for computing the underlying factors characterizing datasets of different types. The latent representation of autoencoders have been studied in the context of enabling interpolation between data points by decoding convex combinations of latent vectors. This interpolation, however, often leads to artifacts or produces unrealistic results during reconstruction. We argue that these incongruities are due to the structure of the latent space and because such naively interpolated latent vectors deviate from the data manifold. In this paper, we propose a regularization technique that shapes the latent representation to follow a manifold that is consistent with the training images and that drives the manifold to be smooth and locally convex. This regularization not only enables faithful interpolation between data points, as we show herein, but can also be used as a general regularization technique to avoid overfitting or to produce new samples for data augmentation.

## 1 INTRODUCTION

Given a set of data points, data interpolation or extrapolation aims at predicting novel data points between given samples (interpolation) or predicting novel data outside the sample range (extrapolation). Faithful data interpolation between sampled data can be seen as a measure of the generalization capacity of a learning system (Berthelot et al., 2018). In the context of computer vision and computer graphics, data interpolation may refer to generating novel views of an object between two given views or predicting in-between animated frames from key frames.

Interpolation that produces novel views of a scene requires input such as the geometric and photometric parameters of existing objects, camera parameters and additional scene components, such as lighting and the reflective characteristics of nearby objects. Unfortunately, these characteristics are not always available or are difficult to extract in real-world scenarios. Thus, in such cases, we can apply *data-driven interpolation* that is deduced based on a sampled dataset drawn from the scene taken under various acquisition parameters.

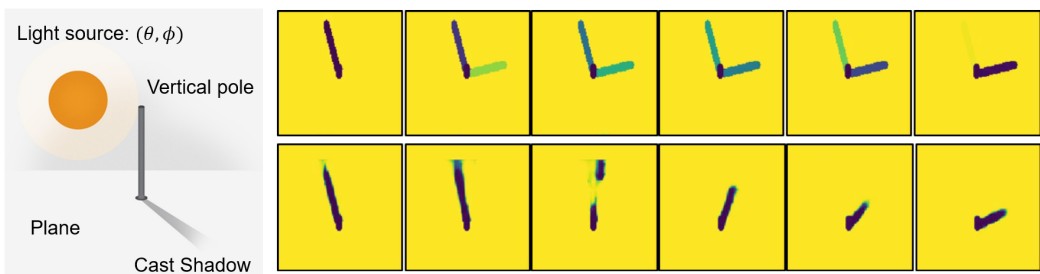

Figure 1: Left: A vertical pole casting a shadow. Yellow blocks-top row: Cross-dissolve phenomena as a result of linear interpolation in the input space. Yellow blocks-bottom row: Image reconstruction obtained by a linear latent space interpolation of an autoencoder. Unrealistic artifacts are introduced.

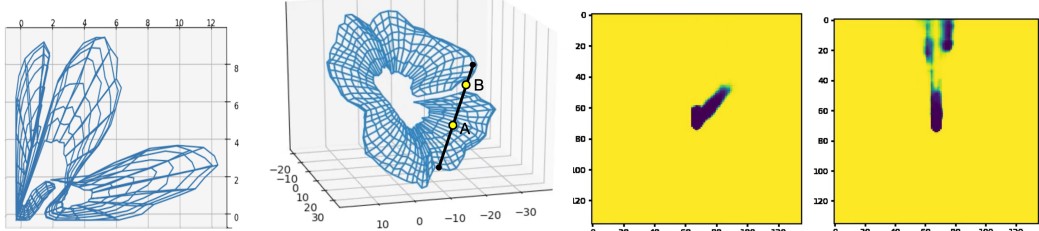

Figure 2: The latent manifold of the data embedded in 2D latent space (leftmost plot) and 3D latent space (second plot from the left) learned by vanilla autoencoders. Gridlines represent the $(\theta, \phi)$ parameterization. The second image from the right was generated from the latent point denoted 'A'. The rightmost image was generated from the latent point denoted 'B'.

The task of data interpolation is to extract new samples (possibly continuous) between known data samples. Clearly, linear interpolation between two images in the input (image) domain does not work as it produces a cross-dissolve effect between the intensities of the two images. Adopting the manifold view of data (Goodfellow et al., 2016; Verma et al., 2018; Bengio et al., 2013), this task can be seen as sampling new data points along the geodesic path between the given points. The problem is that this manifold is unknown in advance and one has to approximate it from the given data. Alternatively, adopting the probabilistic perspective, interpolation can be viewed as drawing samples from highly probable areas in the data space.

One fascinating property of unsupervised learning is the network's ability to reveal the underlying factors controlling a given dataset. Autoencoders (Doersch, 2016; Kingma & Welling, 2013; Goodfellow et al., 2016; Kramer, 1991; Vincent et al., 2010) represent an effective approach for exposing these factors. Researchers have demonstrated the ability to interpolate between data points by decoding a convex sum of latent vectors (Shu et al., 2018; Mathieu et al., 2016); however, this interpolation often incorporates visible artifacts during reconstruction.

To illustrate the problem, consider the following example: A scene is composed of a vertical pole at the center of a flat plane (Figure 1-left). A single light source illuminates the scene and accordingly, the pole projects a shadow onto the plane. The position of the light source can vary along the upper hemisphere. Hence, the underlying parameters controlling the generated scene are $(\theta, \phi)$, the elevation and azimuth, respectively. The interaction between the light and the pole produces a cast shadow whose direction and length are determined by the light direction. A set of images of this scene is acquired from a fixed viewing position (from above) with various lighting directions. Our goal in this example is to train a model that is capable of interpolating between two given images. Figure 1, top row, depicts a set of interpolated images, between the source image (left image) and the target image (right image), where the interpolation is performed in the input domain. As illustrated, the interpolation is not natural as it produces cross-dissolve effects in image intensities. Training a standard autoencoder and applying linear interpolation in its latent space generates images that are much more realistic (Figure 1, bottom row). Nevertheless, this interpolation is not perfect as visible artifacts occur in the interpolated images. The source of these artifacts can be investigated by closely inspecting the 2D manifold embedded in the latent space.

Figure 2 shows two manifolds embedded in latent spaces, one with data embedded in 2D latent space (left plot) and one with data embedded in 3D latent space (2nd plot from the left). In both cases, the manifolds are 2D and are generated using vanilla autoencoders. The grid lines represent the $(\theta, \phi)$ parameterization. It can be seen that the encoders produce non-smooth and non-convex surfaces in 2D as well as in 3D. Thus, linear interpolation between two data points inevitably produces in-between points outside of the manifold. In practice, the decoded images of such points are unpredictable and may produce non-realistic artifacts. This issue is demonstrated in the two right images in Figure 2. When the interpolated point is on the manifold (an empty circle denoted 'A'), a faithful image is generated by the decoder (2nd image from the right). When the interpolated point departs from the manifold (the circle denoted 'B'), the resulting image is unpredictable (right image).

In this paper, we argue that the common statistical view of autoencoders is not appropriate when dealing with data that have been generated from continuous factors. Alternatively, the manifold structure of continuous data must be considered, taking into account the geometry and shape of the manifold. Accordingly, we propose a new interpolation regularization mechanism consisting

Figure 3: Data interpolation using autoencoders. Two points $x_i, x_j$ are located on the input data manifold (solid black line). The encoder $f(x)$ maps input points into the latent space $z_i$, $z_j$ (red arrows). Linear interpolation in the latent space is represented by the blue dashed line. The interpolated latent codes are mapped back into the input space by the decoder $g(z)$ (blue arrows). See Section 2.2 for the contribution of each loss component for an admissible interpolation.

of an *adversarial loss*, a *cycle-consistency loss*, and a *smoothness loss*. The adversarial loss drives the interpolated points to look realistic as it is optimized against a discriminator that learns to tell apart real from interpolated data points. The cycle-consistency and the smoothness losses encourage smooth interpolations between data points. We show empirically that these combined losses prompt the autoencoder to produce realistic and smooth interpolations while providing a convex latent manifold with a bijective mapping between the input and the latent manifolds. This regularization mechanism not only enables faithful interpolation between data points, but can also be used as a general regularization technique to avoid overfitting or to produce new samples for data augmentation, as suggested, among others, by Zhang et al. (2018).

To conclude, the contributions of the papers are: I. We define what constitutes an admissible interpolation between two data points on a continuous manifold. In particular we added the cycle-consistency and the smoothness terms and show their importance in generating admissible interpolations. II. We empirically demonstrate how the combination of the four losses; the reconstruction, adversarial, cycle-consistency and the smoothness losses, contribute to admissible interpolations and produce state of the art results.

## 2 MANIFOLD DATA INTERPOLATION

Before presenting the proposed approach we would like to define what constitutes a proper interpolation between two data points. There are many possible paths between two points on the manifold. Even if we require the interpolations to be on a geodesic path, there might be infinitely many such paths between two points. Therefore, we relax the geodesic requirement and define less restrictive conditions. Formally, assume we are given a dataset sampled from a target domain $\mathcal{X}$. We are interested in interpolating between two data points $x_i$ and $x_j$ from $\mathcal{X}$. Let the interpolated points be $\hat{x}_{i \to j}(\alpha)$ for $\alpha \in [0, 1]$ and let $P(x)$ be the probability that a data point $x$ belongs to $\mathcal{X}$. We define an interpolation to be an *admissible interpolation* if $\hat{x}_{i \to j}(\alpha)$ satisfies the following conditions:

1. **Boundary conditions**: $\hat{x}_{i \to j}(0) = x_i$ and $\hat{x}_{i \to j}(1) = x_j$.

2. **Monotonicity**: We require that under some defined distance on the manifold $d(x, x')$, the interpolated points will depart from $x_i$ and approach $x_j$, as the parameterization $\alpha$ goes from 0 to 1. Namely, $\forall \alpha' \geq \alpha$,

$$d(\hat{x}_{i \to j}(\alpha), x_i) \leq d(\hat{x}_{i \to j}(\alpha'), x_i)$$

and similarly:

$$d(\hat{x}_{i \to j}(\alpha'), x_j) \leq d(\hat{x}_{i \to j}(\alpha), x_j)$$

3. **Smoothness**: The interpolation function $\hat{x}_{i \to j}(\alpha)$ is Lipschitz continuous with a constant K:

$$\|\hat{x}_{i \to j}(\alpha), \hat{x}_{i \to j}(\alpha + t)\| \leq K|t|$$

4. **Credibility**: $\forall \alpha \in [0, 1]$ We require that it is highly probable that interpolated images, $\hat{x}_{i \to j}(\alpha)$ belong to $\mathcal{X}$. Namely,

$$P(\hat{x}_{i \to j}(\alpha)) \geq 1 - \beta, \quad \text{for some constant } \beta \geq 0$$

## 2.1 PROPOSED APPROACH

Following the above definitions for an admissible interpolation, we propose a new approach, called **Autoencoder Adversarial Interpolation** (AEAI), which shapes the latent space according to the above requirements. The general architecture comprises a standard autoencoder with an encoder, $z = f(x)$, and a decoder $\hat{x} = g(z)$. We also train a discriminator $D(x)$ to differentiate between real and interpolated data points. For pairs of input data points $x_i, x_j$, we linearly interpolate between them in the latent space: $z_{i \to j}(\alpha) = (1 - \alpha)z_i + \alpha z_j$, where $\alpha \in [0, 1]$. The first requirement is that we would like $\hat{x}_{i \to j}(\alpha) = g(z_{i \to j}(\alpha))$ to look real and fool the discriminator $D$. Additionally, we add a cycle-consistency loss that encourages the latent representation of $\hat{x}_{i \to j}(\alpha)$ to be mapped back into $z_{i \to j}(\alpha)$ again; namely, $\hat{z}_{i \to j}(\alpha) = f(g(z_{i \to j}(\alpha)))$ should be similar to $z_{i \to j}(\alpha)$. Finally, we add a smoothness loss that drives the linear parameterization to form a smooth interpolation. Putting everything together we define the loss $\mathcal{L}_{i \to j}$ between pairs $x_i$ and $x_j$ as follows:

$$\mathcal{L}^{i \to j} = \mathcal{L}_R^{i \to j} + \lambda_A \mathcal{L}_A^{i \to j} + \lambda_C \mathcal{L}_C^{i \to j} + \lambda_S \mathcal{L}_S^{i \to j} \tag{1}$$

where $\mathcal{L}_R, \mathcal{L}_A, \mathcal{L}_C, \mathcal{L}_S$ are the reconstruction, adversarial, cycle, and smoothness losses, respectively. The first term $\mathcal{L}_R$ is a standard reconstruction loss and is calculated for the two endpoints $x_i$ and $x_j$:

$$\mathcal{L}_R^{i \to j} = \mathcal{L}(x_i, \hat{x}_i) + \mathcal{L}(x_j, \hat{x}_j)$$

where $\mathcal{L}(\cdot, \cdot)$ is some loss function between the two images (we used the $L_2$ distance or the perceptual loss (Johnson et al., 2016)) and $\hat{x}_k = g(f(x_k))$. $\mathcal{L}_A$ is the adversarial loss that encourages the network to fool the discriminator so that interpolated images are indistinguishable from the data in the target domain $\mathcal{X}$:

$$\mathcal{L}_A^{i \to j} = \sum_{n=0}^{M} - \log D(\hat{x}_{i \to j}(n/M))$$

where $D(x) \in [0, 1]$ is a discriminator trying to distinguish between images in the training set and the interpolated images. The cycle-consistency loss $\mathcal{L}_C$ encourages the encoder and the decoder to produce a bijective mapping:

$$\mathcal{L}_C^{i \to j} = \sum_{n=0}^{M} \|z_{i \to j}(n/M) - \hat{z}_{i \to j}(n/M)\|^2$$

where $\hat{z}_{i \to j}(\alpha) = f(g(z_{i \to j}(\alpha)))$. The last term $\mathcal{L}_S$ is the smoothness loss encouraging $\hat{x}(\alpha)$ to produce smoothly varying interpolated points between $x_i$ and $x_j$:

$$\mathcal{L}_S^{i \to j} = \sum_{n=0}^{M} \left\| \frac{\partial \hat{x}_{i \to j}(\alpha)}{\partial \alpha} \right\|^2_{\alpha = n/M}$$

where $\|\partial \hat{x}_{i \to j}(\alpha)/\partial \alpha\|^2_{\alpha = \alpha_0}$ means that the derivative it taken at $\alpha = \alpha_0$. The three losses $\mathcal{L}_A$, $\mathcal{L}_C$ and $\mathcal{L}_S$ are accumulated over $M + 1$ sampled points, from $\alpha = 0/M$ up to $\alpha = M/M$. Finally, we sum the $\mathcal{L}^{i \to j}$ loss over many sampled pairs.

In the next section, we explain the motivation for each of the four losses comprising $\mathcal{L}^{i \to j}$ in Equation 1 and describe how these losses promote the four conditions defined in Section 2.

## 2.2 JUSTIFICATION FOR THE PROPOSED APPROACH

Figure 3 illustrates the justification for introducing the four losses. As seen in Plot A in Figure 3, the images $x_i, x_j$, which lie on the data manifold in the image space (solid black curve), are mapped back to the original images thanks to the reconstruction loss $\mathcal{L}_R^{i \to j}$. This loss promotes the *boundary conditions* defined above. The reconstruction loss, however, is not enough as it neither directly affects in-between points in the image space nor the interpolated points in the latent space. Introducing the adversarial loss $\mathcal{L}_A^{i \to j}$ prompts the decoder $g(z_{i \to j}(\alpha))$ to map interpolated latent vectors back into the image manifold (Plot B). Considering the output of the discriminator $D(x)$ as the probability of image $x$ to be in the target domain $\mathcal{X}$ (namely, to be on the image manifold), the adversarial loss promotes the *credibility condition* defined above. As indicated in Plot B, the encoder $f(x)$ (red

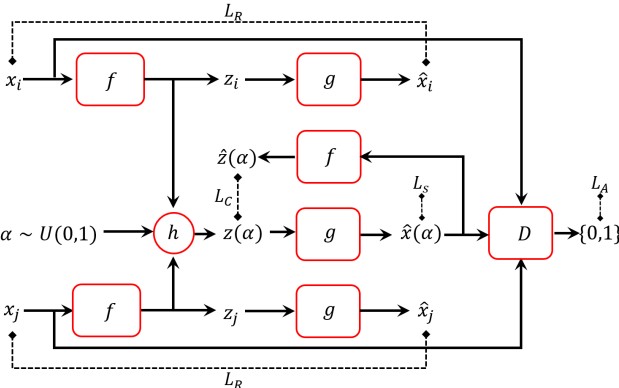

Figure 4: Our proposed architecture. Dotted lines represent the loss functions. $h$ is a non-learned layer that performs latent linear interpolation.

arrows) might, nevertheless, still map in-between images to latent vectors that are distant from the linear line in the latent space. Adding the cycle-consistency loss $\mathcal{L}_C^{i\to j}$ forces the reconstruction of interpolated latent vectors to be mapped back into the original vectors in the latent space (Plot C). The adversarial and cycle-consistency losses encourage bijective mapping (one-to-one and onto) between the input and the latent manifolds, while providing a realistic reconstruction of interpolated latent vectors. Lastly, the parameterization of the interpolated points, namely, $\alpha \in [0, 1]$, does not necessarily provide smooth interpolation in the image space (Plot C); constant velocity interpolation in the parameter $\alpha$ may not generate smooth transitions in the image space. The smoothness loss $\mathcal{L}_S^{i\to j}$ resolves this issue as it requires the distance between $\boldsymbol{x}_i$ and $\boldsymbol{x}_j$ to be evenly distributed along $\alpha \in [0, 1]$ (due to the $L_2$ norm). This loss fulfills the *smoothness condition* defined above (Plot D). If we consider the latent representation as a normed space representing the manifold distance $d(\boldsymbol{x}_i, \boldsymbol{x}_j) = \|\boldsymbol{z}_i - \boldsymbol{z}_j\|$, the linear interpolation in the latent space also satisfies the *monotonicity condition* defined above.

## 2.3 IMPLEMENTATION

The proposed architecture is visualized in Figure 4. At each iteration, we sample two images from our dataset. The two images $(\boldsymbol{x}_i, \boldsymbol{x}_j)$ are encoded by the encoder $f$ into $(\boldsymbol{z}_i, \boldsymbol{z}_j)$, respectively. We sample $\alpha$ uniformly between $[0, 1]$ and pass $(\alpha, \boldsymbol{z}_i, \boldsymbol{z}_j)$ to $h$, a non-learned layer, which calculates the linear interpolation in the latent space, namely, $\boldsymbol{z}_{i\to j}(\alpha) = (1 - \alpha)\boldsymbol{z}_i + \alpha\boldsymbol{z}_j$. We then decode $\boldsymbol{z}_i, \boldsymbol{z}_j$ and calculate the reconstruction loss $\mathcal{L}_R^{i\to j}$. Subsequently, we decode $\boldsymbol{z}_{i\to j}(\alpha)$ and alternately provide the discriminator $D$ with samples either from the training set or from $\hat{\boldsymbol{x}}_{i\to j}(\alpha) = g(\boldsymbol{z}_{i\to j}(\alpha))$. The discriminator is optimized using the standard GAN loss and is updated after every iteration. We then calculate the smoothness loss $\mathcal{L}_S^{i\to j}$ by taking the derivative of $\hat{\boldsymbol{x}}_{i\to j}(\alpha)$ with respect to $\alpha$. Finally, we pass $\hat{\boldsymbol{x}}_{i\to j}(\alpha)$ through the encoder $f$ to obtain $\hat{\boldsymbol{z}}_{i\to j}(\alpha) = f(\hat{\boldsymbol{x}}_{i\to j}(\alpha))$ for the cycle-consistency loss and add the loss $\mathcal{L}_C^{i\to j}(\boldsymbol{z}_{i\to j}(\alpha), \hat{\boldsymbol{z}}_{i\to j}(\alpha))$.

The chosen encoder architecture was VGG-inspired (Simonyan & Zisserman, 2014). We extract the features using convolutional blocks starting from 16 feature maps, gradually increasing the number of feature maps to reach 128 by the last convolutional block. We then flatten the extracted features and pass them through fully connected layers until we reach our desired latent dimensionality. The decoder architecture is symmetrical to that of the encoder. We use max-pooling after each convolutional block and batch normalization with ReLU activations after each learned layer. A random 80%-20% training-testing split was chosen for all experiments, using the same batch size and total number of examples in the dataset. During log grid-search hyperparameter optimization, we found that $\lambda_A = \lambda_C = 10^{-2}$ and $\lambda_S = 10^{-1}$ produce the best results. All experiments were performed using a single NVIDIA V100 GPU.

## 3 RELATED WORK

In its simplest version, the autoencoder (Goodfellow et al., 2016; Kramer, 1991) is trained to obtain a reduced representation of the input, removing data redundancies while revealing the underlying factors of the data set. The reduced space, namely, the latent space, can be viewed as a 'useful'

representation space in which data interpolation can be attempted. Many autoencoder improvements have been proposed in recent years, including new techniques designed for improved convergence and accuracy. Among these are the introduction of new regularization terms, new loss objectives (such as adversarial loss) and new network designs (Doersch, 2016; Kingma & Welling, 2013; Larsen et al., 2015; Makhzani et al., 2015; Vincent et al., 2010; Goodfellow et al., 2016; Vincent et al., 2010). Other new autoencoder techniques provide frameworks that attempt to shape the latent space to be efficient with respect to factor disentanglement or to make it conducive to latent space interpolation (Kingma & Welling, 2013; Bouchacourt et al., 2017; Vincent et al., 2008; Yeh et al., 2016; Higgins et al., 2016).

Within this second category, the variational autoencoder (VAE) and its derivatives were shown to be very successful in applying interpolation in the latent space, in particular for multimodal distributions, such as MNIST. The KL term in the VAE loss tends to cluster the modes in the latent space close to each other (Dieng et al., 2019). Consequently, linearly interpolating between different modes in the latent space may provide pleasing results that smoothly transition between the modes. Unfortunately, this cannot be applied to data points whose generating factors are continuous (in contrast to multimodal distributions) given that the KL loss term tends to fold the manifold tightly into a compact space making it highly non-convex.

Berthelot et al. (2018) propose using a critic network to predict the interpolation parameter $\alpha \in [0, 1]$ while an autoencoder is trained to fool the critic. The motivation behind this approach is that the interpolation parameter $\alpha$ can be estimated for badly-interpolated images, while it is unpredictable for faithful interpolation. While this approach might work for multimodal data, it does not seem to work for data sampled from a continuous manifold. In such cases, the artifacts and the unrealistic-generated data do not provide any hint about the interpolating factor.

The GAIA method of Sainburg et al. (2018) uses BEGAN architecture composed of a generator and a discriminator, both based on autoencoders. The discriminator is trained to minimize the pixel-wise loss of real data and to maximize the pixel-wise loss of generated data (including interpolations). On the other hand, the generator is trained to minimize the loss of the discriminator for the interpolated data. The GAIA method is devoted to synthesizing realistic-looking images while ignoring the objective of image diversity and the need for smooth transitions between data points.

Perhaps the method most similar to our approach is the *adversarial mixup resynthesis* (AMR) of Beckham et al. (2019). With the AMR method, a decoded mixup of latent codes $Mix(\boldsymbol{z}_i, \boldsymbol{z}_j)$ are encouraged to be indistinguishable from real samples by fooling a trained discriminator. This is similar to the adversarial loss introduced in our framework. Nevertheless, as elaborated in Section 2.2 and illustrated in Figure 3 (Plot B), the adversarial loss alone only amounts to generating realistic-looking interpolations, where the latent space is prone to mode collapse and sharp transitions along the interpolation paths.

In contrast to these methods, our additional smoothness and cycle-consistency requirements not only generate smooth transitions between data points but also ensure a diverse generation of realistic-looking images while avoiding mode collapse and sharp transitions along the interpolating paths. This characteristic will be demonstrated in Section 4 and in the ablation study provided in the appendix.

## 4 RESULTS

Evaluating the realisticality of interpolation is often elusive. In the unsupervised scenario, where the ground-truth parameterization is unavailable, defining a path between two points $\boldsymbol{p}_i, \boldsymbol{p}_j$ in the parameter space depends on the parameterization of the underlying factors governing the data, which is unknown. For example, in our synthetic pole dataset, the parameter space is $(\theta, \phi)$ and there are infinitely many possible paths between any two points in that space, each of which can yield an admissible interpolation. Nevertheless, we evaluate the interpolation faithfulness both qualitatively and quantitatively on various datasets based on the conditions we defined in Section 2.

### 4.1 DATASET

We tested our method against two different datasets: the synthetic pole dataset, which was rendered using the *Unity* game engine, where all images were taken from a fixed viewing position (from above)

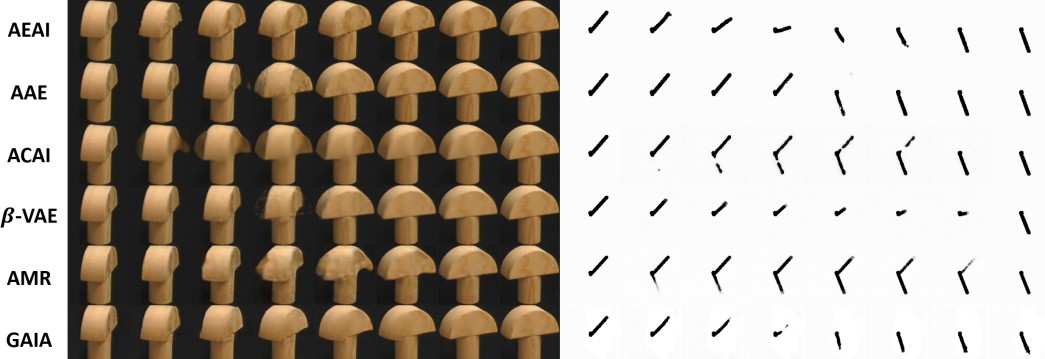

Figure 5: Each of the six rows presents linear interpolation of images from COIL-100 and our synthetic dataset for each of the methods tested.

and the COIL-100 dataset. For the first dataset, a single illumination source was rotated at intervals of 5 degrees along the azimuth at different altitudes, ranging from 45 to 80 degrees with respect to the plane in 5-degree intervals. This dataset contains a total of 576 images. In the second dataset, to test our method against real images with complex geometric and photometric parameterization, we used the COIL-100 dataset (Nene et al., 1996) containing color images of 100 objects. The objects were placed on a motorized turntable against a black background. The images were taken at intervals of 5 degrees resulting in a total of 72 images for each class. All architectures were trained using a dataset that contained a single class from COIL-100 or the synthetic pole dataset.

## 4.2 QUALITATIVE ASSESSMENTS

Each one of the six rows in Figure 5 presents a linear interpolation of a single object from the COIL-100 dataset (left) and our pole dataset (right) sampled from the validation dataset. We compared the results of the $\beta$-Variational Autoencoder ($\beta$-VAE) (Higgins et al., 2016), the Adversarial Autoencoder (AAE) (Makhzani et al., 2015), the Adversarially Constrained Autoencoder Interpolation (ACAI) (Berthelot et al., 2018), the Generative Adversarial Interpolative Autoencoding method (GAIA) (Sainburg et al., 2018), the Adversarial Mixup Resynthesis (AMR) (Beckham et al., 2019) and our approach–Autoencoder Adversarial Interpolation (AEAI). In the experiments with both datasets, we used a latent dimensionality of 256. From Figure 5 it can be seen that our proposed method provides realistic-looking reconstructions and an admissible interpolation between modes. The AAE and $\beta$-VAE interpolations change abruptly between modes and introduce small artifacts during reconstruction. The ACAI produces unrealistic transitions and artifacts during reconstruction, especially in the mid-range of the $\alpha$-values. The GAIA method produces realistic transitions with small artifacts during reconstruction while the AMR produces significant artifacts and gradual transitions between modes. Additional qualitative results and an ablation study are presented in the Appendix.

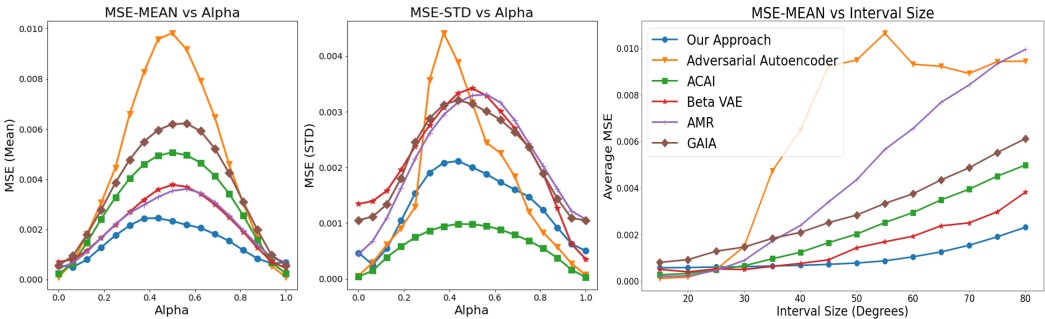

Figure 6: We use the parameterization of the dataset to evaluate the reconstruction accuracy of the AAE, ACAI, $\beta$-VAE, AMR, GAIA and our proposed method. Left graph: Averaged MSE vs. $\alpha$ values. Middle graph: STD of MSE vs. $\alpha$ values. Right: Averaged MSE of the interpolated images vs. the interval length.

### 4.3 QUANTITATIVE ASSESSMENTS

For a quantitative comparison we used the COIL-100 dataset. We fixed an interval length, which is a multiplicative of 5 degrees, and calculated the reconstruction error (MSE) against the available ground-truth images. We used an interval length of 80 degrees that resulted in 14 in-between images of a single object. The reconstruction error of the interpolated images is presented in Figure 6. Clearly, our method reduces the mean MSE and the standard deviation of the MSE for different alpha values. We then inspected the average reconstruction error on multiple intervals ranging from 15 to 80 degrees as presented in the right part of Figure 6. Note that our proposed method is able to reduce the reconstruction error of interpolated images consistently even when the interval length increases.

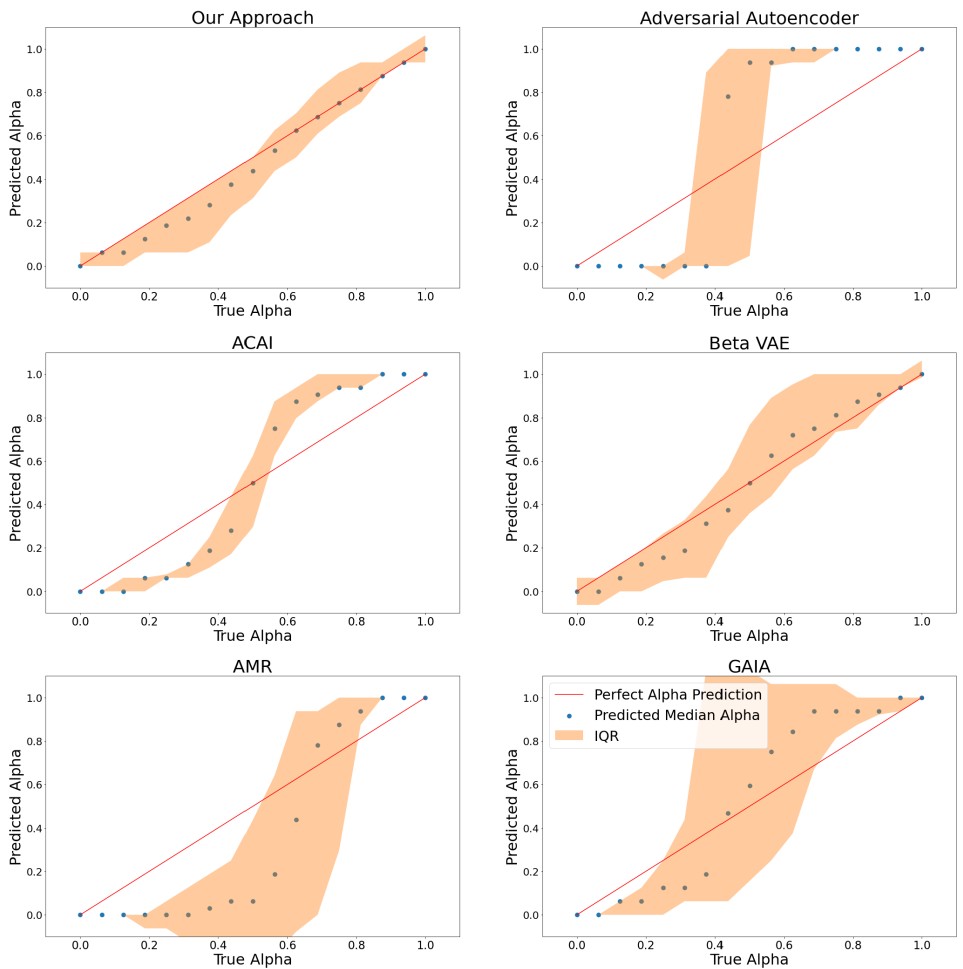

Figure 7: Predicting the interpolated alpha value based on the $L_2$ distance of the interpolated image to the closest image in the dataset. The dots represent the median and the colored area corresponds to the interquartile range.

To assess the transition smoothness from one sample to the other, we compared each interpolated image $\hat{x}_{i \to j}(\alpha)$ to the closest image in the dataset in terms of the $L_2$ distance and assigned the alpha value for the interpolated image according to the retrieved image. We repeated this process for all the intervals of length 70. Figure 7 presents the scatter diagrams for each method. It is demonstrated that our framework consistently retrieves the best value of alpha with a smaller interquartile range (IQR). The next experiment was applied to the synthesized pole dataset. As above, we retrieved the closest image in terms of MSE in the image space, and measured the $L_2$ distance in the *parameter space* between the interpolated image and the source image ($\alpha = 0$) and between the interpolated image and the target image ($\alpha = 1$). We repeated this process on multiple intervals of different lengths on both $\theta$ and $\phi$, and present the average distance from the source and target images as a function of

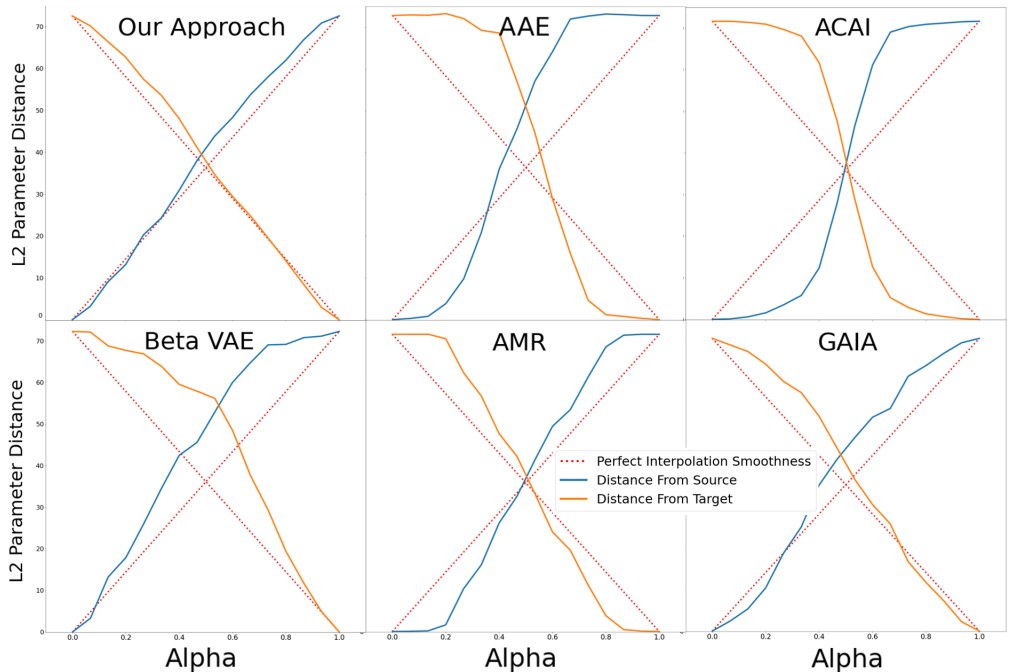

Figure 8: We sampled two images $x_i, x_j$ and linearly interpolated between them in latent space. For each interpolated image, we retrieved the closest image in terms of MSE from the dataset. The blue and orange lines present the averaged $L_2$ distance, in the parameter space $(\theta, \phi)$, between the retrieved image and $x_i, x_j$, respectively. The red lines represent perfect interpolation smoothness.

the interpolation variable, $\alpha$. Figure 8 shows the results for each tested method. It is demonstrated that all methods exhibit monotonicity characteristics; however our approach outperforms the other methods with respect to the smoothness of the parameterization. Our approach tightly follows the linear parameterization due to the explicit smoothness term we incorporated into the training scheme.

## 5 CONCLUSION & DISCUSSION

The problem of realistic and faithful interpolation in the latent spaces of generative models has been tackled successfully in the last few years. Nevertheless, it is our opinion that generative approaches that deal with manifold data are not as common as multimodal data, and this misinterpretation of manifold data harms the competence of generative models to deal with them successfully. In this work, we argue that the manifold structures of data generated from continuous factors should be taken into account. Our main contribution is applying convexity regularization using adversarial and cycle-consistency losses. Applying this technique on small datasets of images, taken from various viewing conditions, we managed to greatly improve the fidelity of interpolated images. We also implemented a smoothness loss and improved the non-uniform parameterization of the latent manifold. In future work, we intend to further investigate properties of latent manifolds, in particular, capable of generating admissible interpolation between both categorized and continues data, and use the proposed approach as a general regularizer method for generative models with few training examples.

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

APPENDIX

# A   ADDITIONAL RESULTS

In this section, we provide additional results showing the interpolation behavior of different architectures we studied. Each of the following figures is composed of six blocks, each of which presents a bilinear interpolation of the latent representation of four ground truth images that reside in each corner of the block. Figures 9 and 10 demonstrate bilinear interpolation on a differnet COIL-100 objects and Figure 11 shows the interpolation results on our synthetic pole dataset.

We demonstrate that our technique produces admissible interpolation while other techniques fail to reconstruct in-between images realistically or to transition smoothly from mode to mode. For example, in Figure 11 we show that the AMR, AAE and the ACAI create a cross-dissolve effect during interpolation while the interpolation of the $\beta$-VAE and GAIA method changes abruptly while showing little progression between interpolation frames. Additionally, in Figures 9 and 10 we show that the AAE and AMR exhibit artifacts in interpolated images while the ACAI shows blurry images during reconstruction. The $\beta$-VAE and GAIA shows mostly realistic reconstructions; however, the transition is not smooth nor consistent.

# B   ABLATION STUDY

We present an ablation study of our unsupervised interpolation framework presented in Figure 4 above. As seen in the bottom left part of Figures 12 and 13, without a significant contribution from the discriminator, the reconstructed images resulting from interpolating latent vectors are unrealistic and exhibit severe artifacts and non-smooth transition between modes. Without the cycle-consistency loss, interpolated images are relatively realistic; however, they change modes abruptly, exhibiting artifacts when transitioning from one mode to another. Adding both cycle consistency and the discriminator results in realistic transitions from mode to mode as can be seen in the bottom right part of Figure 12. Nevertheless, there are instances of consecutive interpolated images that show little to no change as can be seen in the two leftmost columns of the bottom right part of Figure 12. When introducing the smoothness loss, as can be seen in the top left part of Figure 12, we get both smooth transitions and realistic reconstructions of interpolated latent vectors.

We present a quantitative analysis of our ablation study in Figures 14 and 15. For each case, we demonstrate the average MSE error between the interpolated image and ground truth images retrieved from our dataset. In the top part of Figure 14, we fix an interval length of 80 degrees and iterate over all such intervals in our dataset. We split the interval into 16 images separated by 5 degrees and obtain 14 in-between images. For every interpolated image we retrieve the corresponding ground truth image in our dataset and present the average MSE and standard deviation on all such intervals. In the bottom part of Figure 14 we repeat this process on multiple interval sizes ranging from 15 degrees to 80 degrees. We show that each element in our loss function 1 contributes to reducing the mean and variance of the reconstruction error. We further support our justification presented in Figure 3 by showing that the largest contribution to the reduction of the MSE stems from the introduction of the discriminator, which keeps the reconstruction of interpolated images consistent with the dataset. The addition of the cycle-consistency and smoothness losses further improves the results by encouraging a smooth bijective mapping.

In Figure 15 we present the results of predicting the interpolated alpha value by querying the dataset for the closest image in terms of the $L_2$ distance to the interpolated image. We present the median for each alpha on all intervals with the corresponding interquartile range. The red line demonstrates the perfect retrieval of the predicted alpha value. It is shown that the absence of the discriminator greatly affects the interpolation faithfulness while the addition of the cycle-consistency and smoothness losses contributes to the consistency of retrieving the correct alpha value.

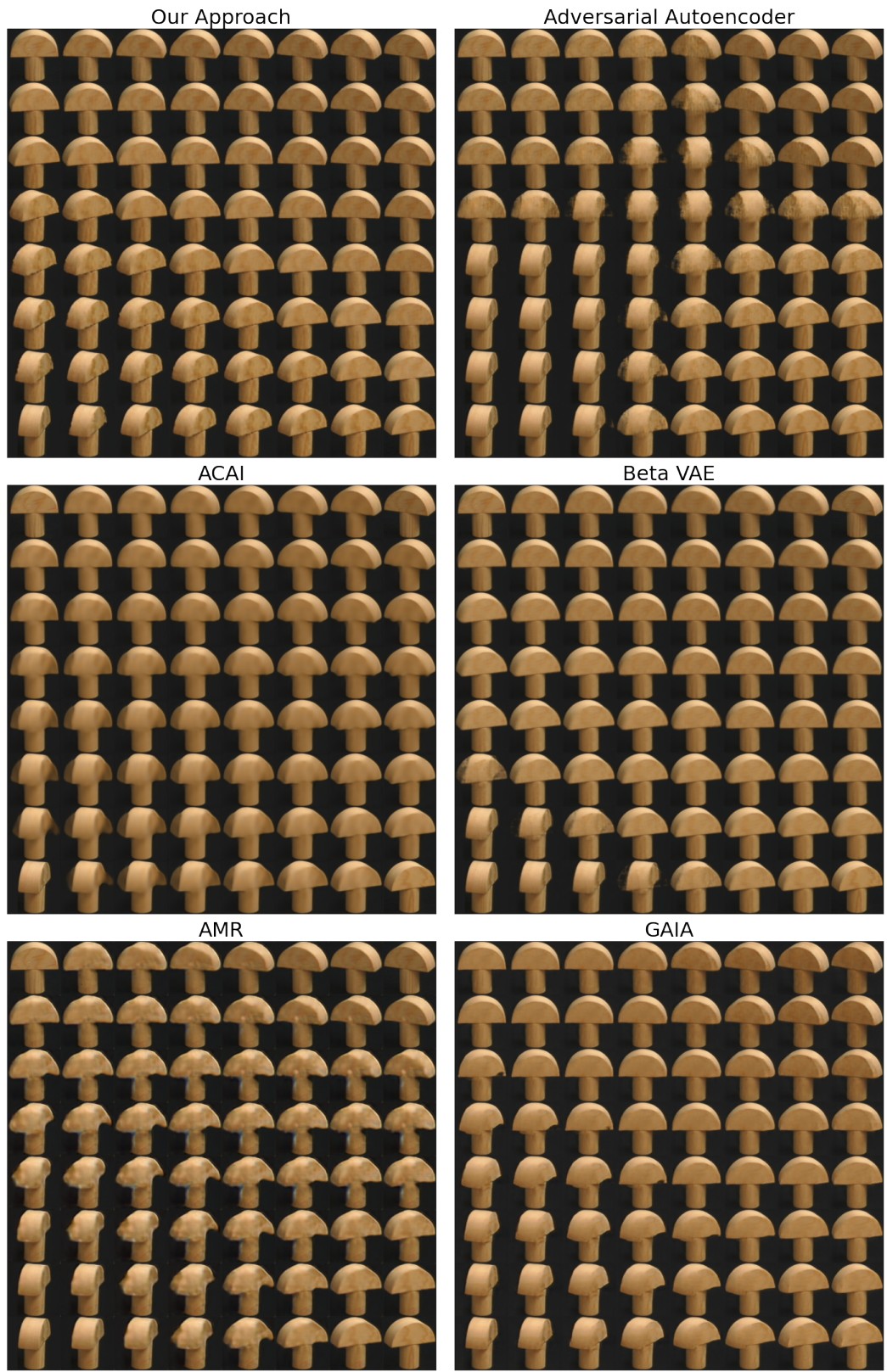

Figure 9: Each of the six blocks shows a bilinear interpolation of four ground truth images that reside in each corner of the block.

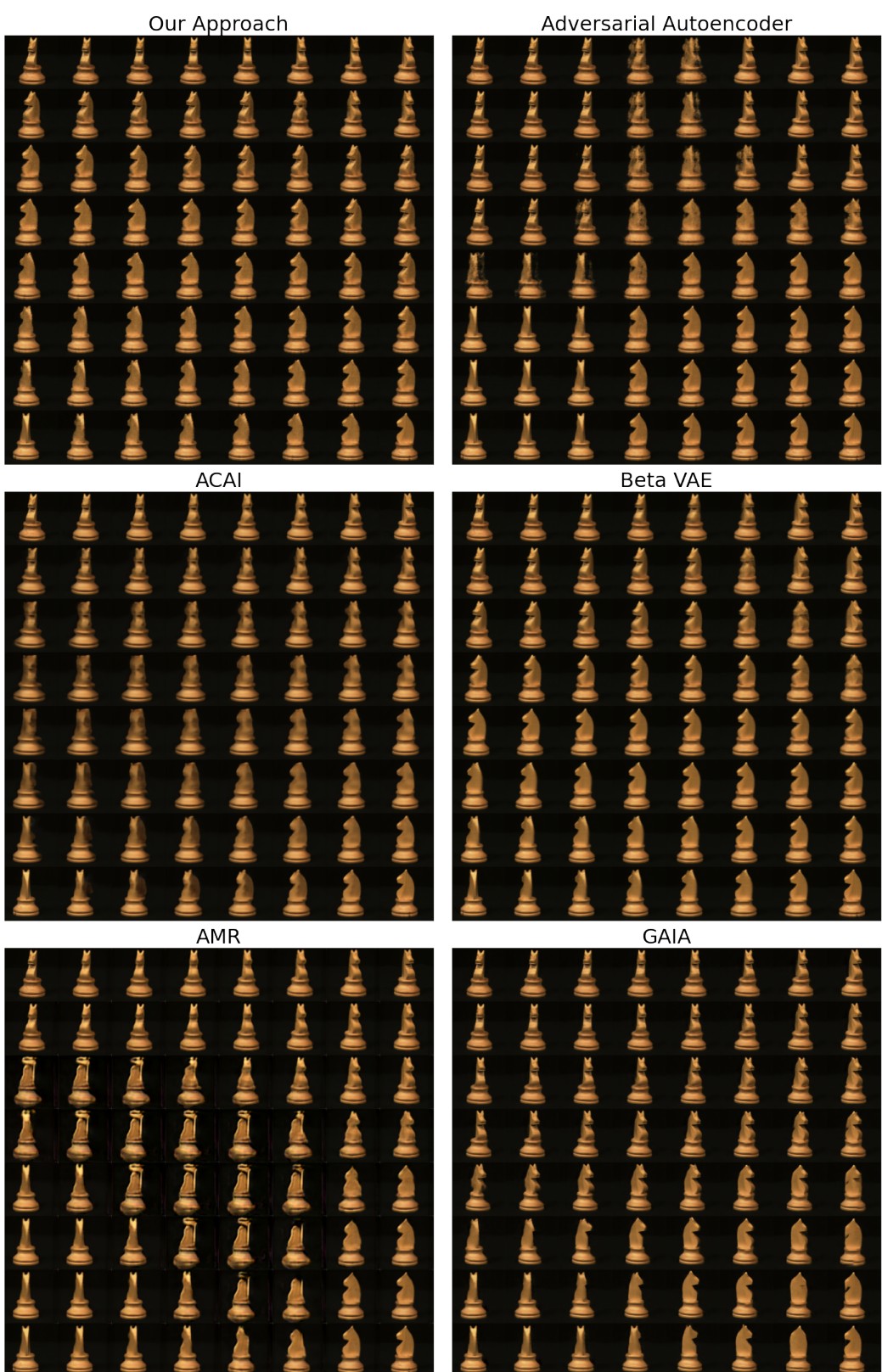

Figure 10: Each of the six blocks shows a bilinear interpolation of four ground truth images that reside in each corner of the block.

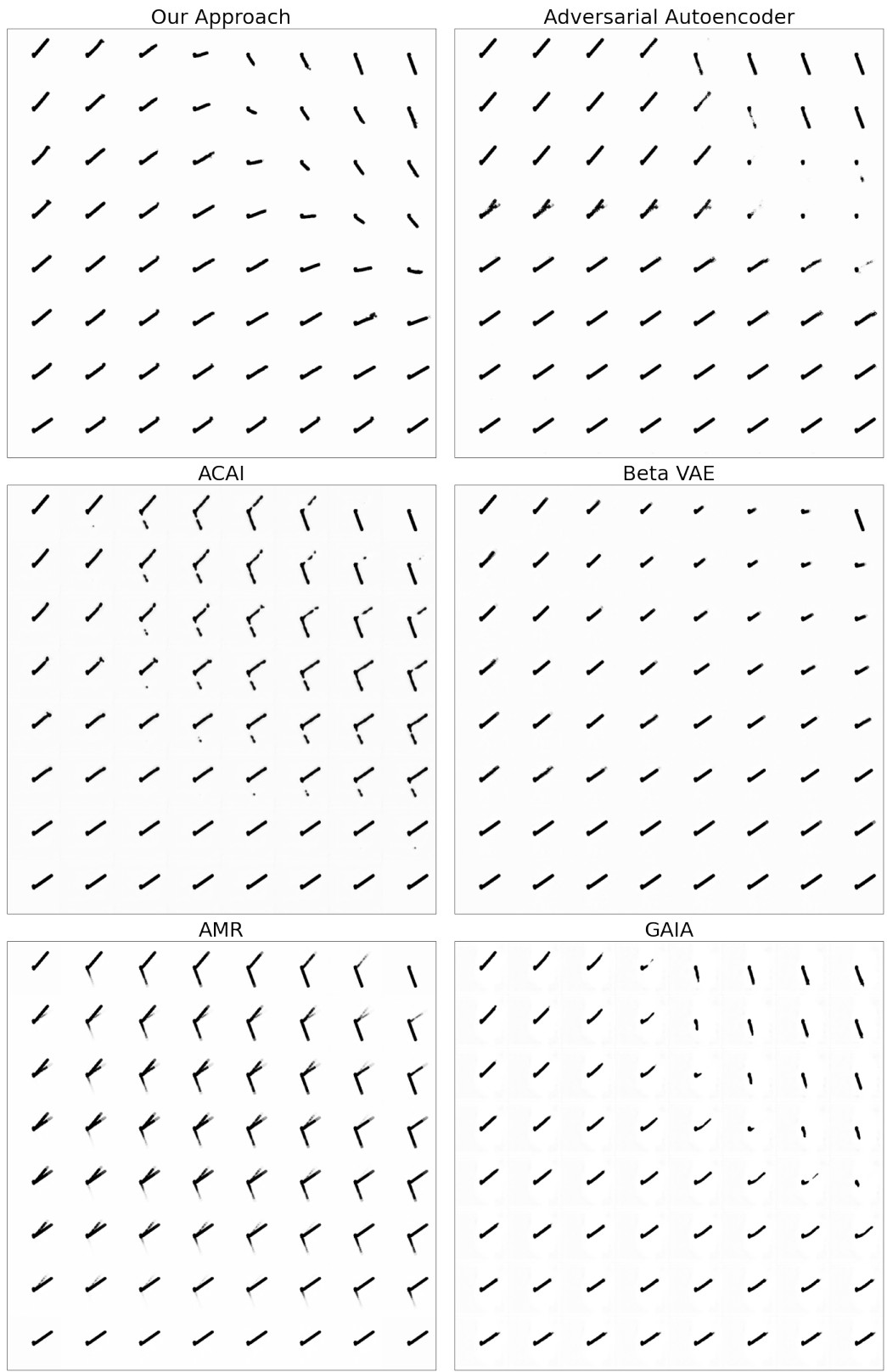

Figure 11: Each of the six blocks shows a bilinear interpolation of four ground truth images that reside in each corner of the block.

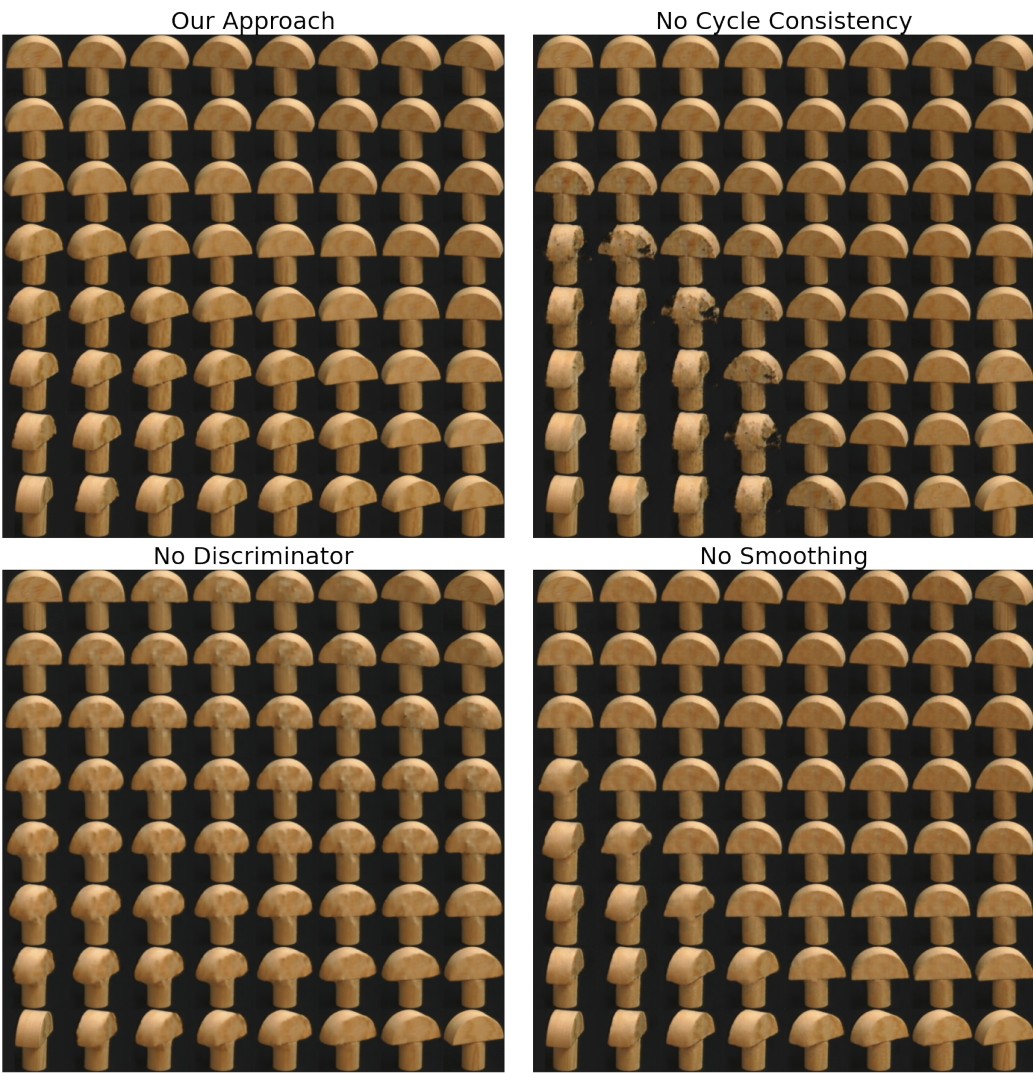

Figure 12: Each of the four blocks presents bilinear interpolation of four ground truth images that reside in each corner of the block. Top left: Bilinear interpolation results of our approach with all loss components. Top right: Removing the cycle-consistency contribution from the loss function. Bottom left: Removing the discriminator contribution. Bottom right: Removing the smoothing contribution.

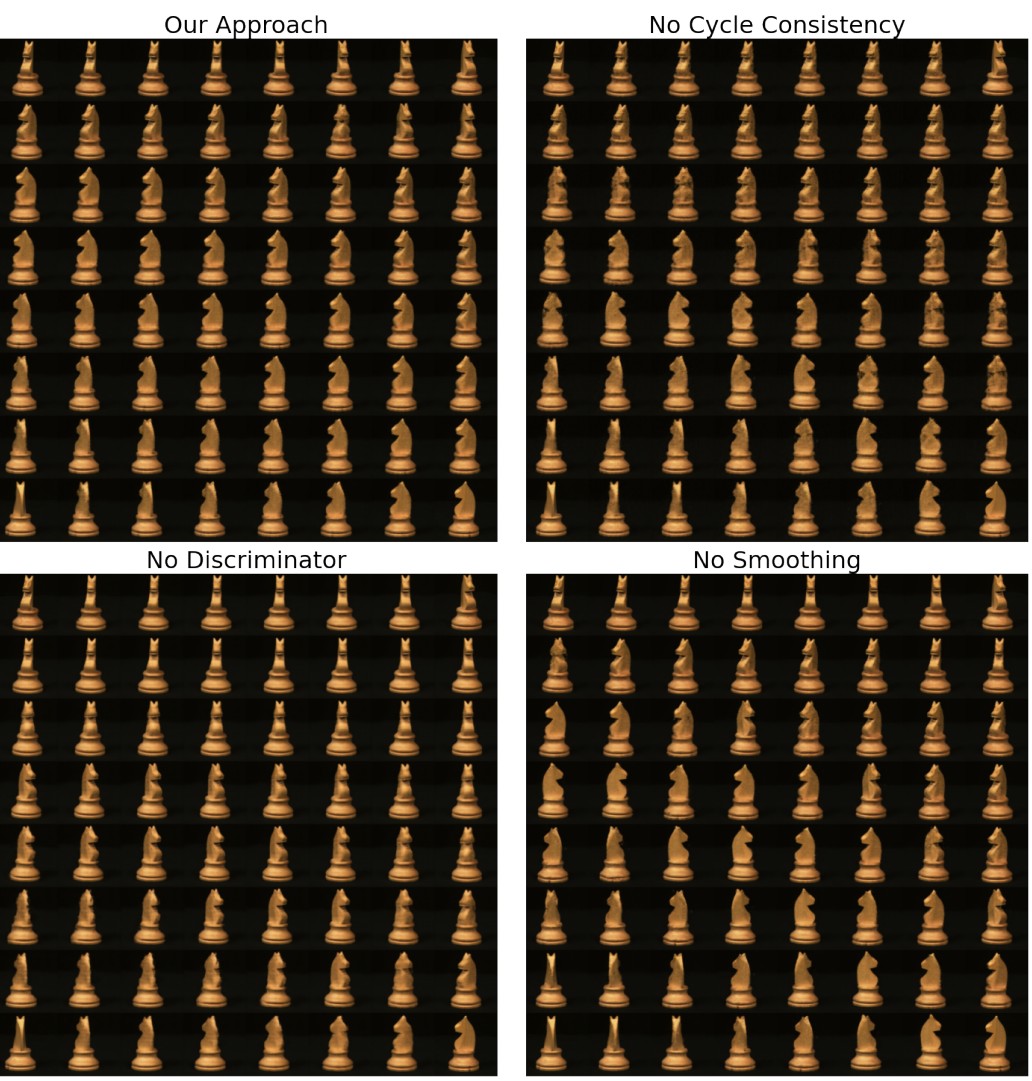

Figure 13: Each of the four blocks presents bilinear interpolation of four ground truth images that reside in each corner of the block. Top left: Bilinear interpolation results of our approach with all loss components. Top right: Removing the cycle-consistency contribution from the loss function. Bottom left: Removing the discriminator contribution. Bottom right: Removing the smoothing contribution.

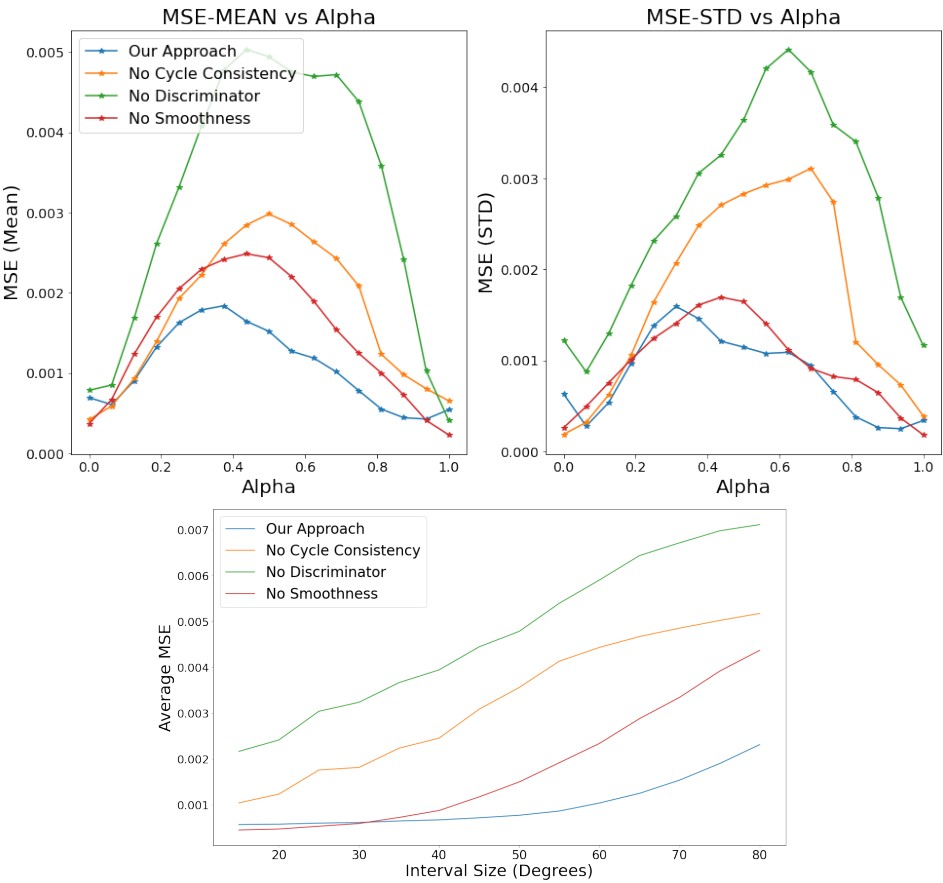

Figure 14: Top graph: Average reconstruction error and standard deviation vs. $\alpha$ values. Bottom: Average MSE of the interpolated images vs. the interval length.

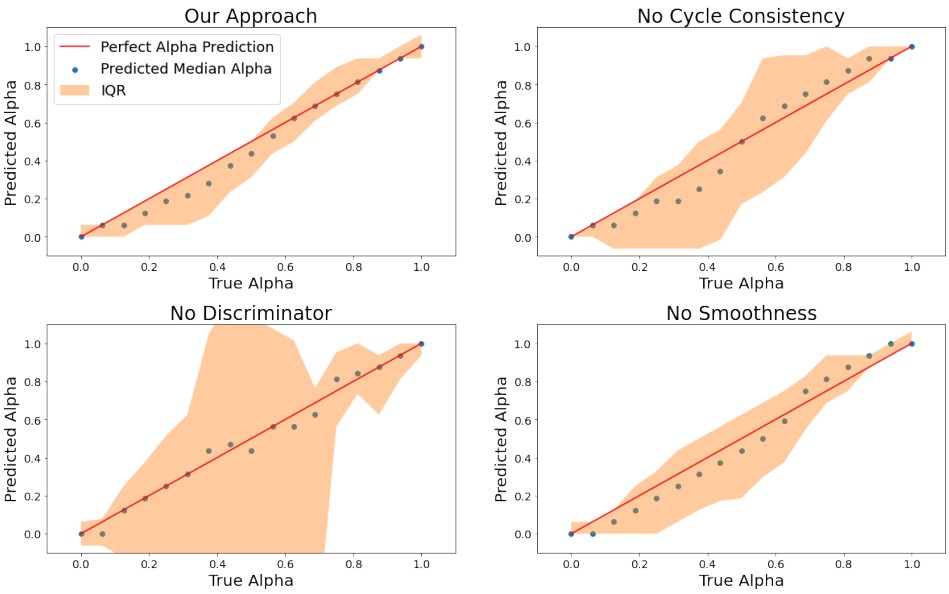

Figure 15: Predicting the interpolated alpha value based on the $L_2$ distance of the interpolated image to the closest image in the dataset. The dots represent the median and the colored area corresponds to the interquartile range.

