# OpenReview forum: "Autoencoder Image Interpolation by Shaping the Latent Space"
_ICLR.cc/2021/Conference — Reject_

### Official Review · AnonReviewer3 · 2020-10-26
**Nice paper, a bit limited contribution**

**Rating:** 6
**Confidence:** 4

**Review:**

This paper introduces several autoencoder (AE) regularization terms that aim at reproducing continuous realistic deformation by interpolating latent codes of images. The authors assume there is a continuous process generating the data and introduce three novel loss terms (in addition to the standard AE reconstruction loss). The first term is a GAN loss for decoded interpolated latents $\hat{x}(\alpha)$ where this terms makes sure the interpolated latents are decoded to images similar to the train images. The second term is called cycle-consistency and enforce injectivity of the decoder. The last term is enforcing smoothness of the decoded image as a function of the interpolated latent. Combining these three losses with the original loss leads to natural interpolations of latents that enjoy both smoothness and realism.  The method is tested on a synthetic "pole shadow" example, and COIL 100. The method seem to improve upon several baselines on these datasets.

More details:

- The method is simple and able to improve latent interpolations. I think the choice of losses and properties they enforce is well explained.  The results are convincing and seem to provide a good arrangement of the latent space.
- In terms of contributions: the incorporation of a discriminator loss in training of AE to provide more realistic interpolations was done before, as the authors acknowledge (e.g., Beckham et al. 2019). This diminish some of this paper's contribution. However, as shown in Figure 12, discriminator loss alone does not solve the continuous latent problem.
- In Figure 12, I cannot see the benefit in the smoothing term, that is, top-left and bottom-right images look almost identical to me. Can you state if there is some difference?  If not, did you encounter a benefit in incorporating the smoothness term somewhere else?
- Are the corners in the square interpolation grids (e.g., Figures  5, 9, 10, 11, 12, 13) train or test examples? How does grid interpolation of test examples look? The last question is interesting both for in-distribution test examples, as well as out-of-distribution test examples.
- How does the discriminator $D$ is trained? Do you use standard GAN loss? Missing info here.
- How do you set the different parameters $\lambda_i$ of the model?
- How is the loss $\mathcal{L}_S$ approximated? Do you use automatic differentiation, numerical differentiation? Do you use stochastic approximation?
- Figure 8 was not clear to me.
- Why L1 distance of images is used in 4.2 and L2 in 4.3?
- Minor: typo "qualitatively and qualitatively" ; distorted text in Figure 6

---

> ### Author Response · Authors · 2020-11-23
> **Response to Reviewer 3**
>
> **Q3.1**: In Figure 12, I cannot see the benefit in the smoothing term…Can you state if there is some difference? If not, did you encounter a benefit in incorporating the smoothness term somewhere else?
>
> **A3.1**: We have included additional examples of the different methods where the difference is clearly visible. In addition, the included quantitative results also support the addition of the smoothing term.
>
> **Q3.2**: Are the corners in the square interpolation grids train or test examples? How does grid interpolation of test examples look?
>
> **A3.2**: All interpolation grids are created by sampling the test dataset. We made it clear in the paper.
>
> **Q3.3**: How does the discriminator D is trained? Do you use standard GAN loss? Missing info here.
>
> **A3.3**: We are using standard GAN loss where the discriminator is updated at every iteration. We made it clear in the paper.
>
> **Q3.4**: How do you set the different parameters $\lambda_i$ of the model?
>
> **A3.4**: We used hyperparameter tunning using log grid-search. We made it clear in the paper.
>
> **Q3.5**: How is the loss Ls approximated? Do you use automatic differentiation, numerical  differentiation?
>
> **A3.5**: We apply the analytical derivative (by backpropagating through $\alpha$)
>
> **Q3.6**: Figure 8 was not clear to me.
>
> **A3.6**: Given the synthetic pole dataset, we sampled two points $(x_i, x_j)$ and interpolated between them. For each interpolated image, we retrieved the closest image from the dataset using $L_2$ loss and calculated the distance from the retrieved image to $x_i$ and $x_j$. We demonstrate that our approach transitions smoothly between any two initial points $(x_i, x_j)$. We made it clearer in the paper.
>
> **Q3.7**: Why L1 distance of images is used in 4.2 and L2 in 4.3?
>
> **A3.7**: Typo corrected. We used $L_2$ distance throughout the paper.
>
> **Q3.8**: Minor: typo "qualitatively and qualitatively"; distorted text in Figure 6.
>
> **A3.8**: Typo and visualization corrected.

---

### Official Review · AnonReviewer4 · 2020-10-26
**Clearly presented with compelling results.**

**Rating:** 7
**Confidence:** 4

**Review:**

## Summary
The paper presents a method of regularising the latent space of an Autoencoder in a way that pressures the data manifold to be convex. This allows interpolation within the latent space which does not leave the data manifold and results in realistic reconstructions as one moves from one point to another.

This is done through the introduction of adversarial and cycle-consistency losses, over and above the usual reconstruction loss and a smoothness loss. The adversarial loss ensures that the interpolated reconstructions are realistic, while cycle-consistency encourages a bijective mapping.

## Quality & Clarity

The paper is clearly written and without typographical or grammatical errors. It is structured logically and the authors' arguments are easily followed.

The results are compelling and the proposed AEAI technique clearly outperforms other methods in the qualitative experiments, with quantitative results to substantiate it.

## Originally & Significance

The contribution of the paper is clear in that it imposes convexity regularisation to the latex space. The approach is compared with modern competing techniques and the work is well positioned among recent literature in the field.

## Outcome

This is a clear, high-quality paper with compelling results.

---

> ### Author Response · Authors · 2020-11-23
> **Response to Reviewer 4**
>
> Thank you for your time, effort, and your extremely positive review of our paper.

---

### Official Review · AnonReviewer2 · 2020-10-28
**This paper has the potential to be a great paper but in its current form would really benefit from more results and another pass of writing**

**Rating:** 6
**Confidence:** 4

**Review:**

# Summary
This paper proposes new regularization terms (smoothness and cycle consistency) for more realistic latent interpolations in auto-encoders. It is experimentally tested by measuring interpolation error on two datasets: a new custom pole dataset introduced by the paper and the COIL-100 dataset.

# Pros
1. The method feels intuitively motivated and makes sense.
2. The results appear convincing compared to the (limited choice of) baselines.
3. Ablation study helps comprehend the contribution of each loss term.

# Cons
1. No direct comparison to state of the art baselines, namely GAIA and AMR.
2. Results only on artificial tasks. For example, there’s no measure of the effects of the better conditioned latent space on downstream tasks such as a classification on SVHN or CIFAR10 like some other works in the domain checked.
3. Novelty is not clearly identified: a good place would be the end of section 1, e.g. “Our contributions are …”. I deduced it was the cycle consistency loss (which seems inspired from cycle GANs) and smoothness loss since the other loss terms look like what previous techniques already do.
4. Ablation should really be in the main section of the paper since it’s important. On the other hand Figures (3) and (4) which didn’t really add to my comprehension could be removed to save space.
5. Without the appendix the paper is lacking essential information.
6. The bibliography seems to be lacking (see questions and nits)

# Questions and nits
1. “This regularization ... can also be used as a general regularization technique to avoid overfitting or to produce new samples for data augmentation.” It would be good to see it demonstrated in results.
2. In several places, you cite auto-encoders but really refer to variational auto-encoders, not crediting earlier works.
3. “Researchers have demonstrated the ability to interpolate between data points by decoding a convex sum of latent vectors (Shu et al., 2018) ...”  I believe there’s prior art before that, for example, just to name one: https://arxiv.org/abs/1611.03383
4. “... interpolated points to look reliable as it is optimized ...” In several places the word “reliable” is used, I suppose you meant “realistic”. If not can you clarify what it means?
5. “... while providing a convex latent manifold with a bijective mapping between the input and latent spaces.” It seems counter intuitive to me, the input space is much larger than the latent space, how can it be bijective? In addition you later introduce $L_c$ the cycle consistency loss which tries to map $f(g(z_{i,j,\alpha}))$ to $z_{i,j,\alpha}$, in other words it looks surjective only. I understand that perceptually you want $g(f(x))$ to be close to $x$, but they are not the same, or are they? If you are referring to the mapping from the latent space onto itself by $f(g(.))$, then the term `identity mapping` would seem clearer to me.
6. “Credibility” I don’t understand why it is written in such a complicated manner. Why not simply write $P(\hat{x}_{i,j,\alpha}) \geq 1 - \beta$ for a constant $\beta \geq 0$.
7. Concerning the smoothness loss $L_s$ it seems to be derived from the k-Lipschitz constraint (3). But in its implementation it’s really 0-Lipschitz, it would be great to clarify this aspect.
8. In addition, concerning $L_s$, the formulation seems odd. I’d expect it to be written as $\sum ||\frac{d\hat{x}_{i,j}(\alpha)}{d\alpha}(n/M)||^2$ since in its current form $\alpha$ does not appear in the numerator.
9. One last thing concerning $L_s$, it would be good to give the reader a simple explanation: if I’m correct you’re simply minimizing the pixel distance between successive images on the interpolation line. That would be really helpful to state it clearly - assuming I understood correctly.
10. “2.2 JUSTIFICATION FOR THE PROPOSED APPROACH”. I found it was already justified by the intro to section 2 for the most part. One exception is the justification for the cycle consistency loss L_c which was not covered in the introduction of section 2. So you could simplify/remove this section by moving it to section 2.
11. On the topic of L_c, it is justified as “the cycle-consistency loss L_c forces the encoder-decoder architecture to map linearly interpolated latent vectors onto the image manifold …”. I don’t see why it should. As I understand it, the loss term itself only seems to say that it should map to an image that projects to the same latent representation. This could use clarification.
12. In the comparison to other methods, did you use the same number as training examples? I’m asking this question because the loss term L_s has an implicit batch size of M which could lead your method to see M times more samples than the methods under comparison (in particular AAE and ACAI).
13. “The two images (xi, xj) are encoded by the shared-weight encoder…”. I didn’t follow what the weights are shared with.
14. The loss terms are called $L_{a,c,s}$ but the weighting hyper-parameters are called $\lambda_{1,2,3}$. Using the same letter as their corresponding loss term instead of 1,2,3 would make reading more friendly.
15. “The GAIA method of Sainburg et al. (2018) is similar in spirit to the AMR framework.” The GAIA method came first, so this is the other way around (B is similar in spirit to A, if B came after A). Actually it should probably be presented before AMR.
16. “but also ensure a diverse generation ... while avoiding mode collapse“. How does it ensure it? I didn’t really get what loss terms directly prevented modal collapse nor diversity? Or if the effect is indirect, could you give more insights on what causes it?
17. The results section comes just after Related Work and it’s striking that there’s no comparison to the two methods yours is most similar to and that were covered sentences ago.
   1. I understand you claim that “Comparisons with AMR and GAIA methods (Beckham et al., 2019; Sainburg et al., 2018) are analogous to the ablation study presented in the Appendix, where the smoothness and cycle-consistency losses are missing.” but it’s not clear whether you ran the experiment and observed it was indeed analogous or whether it should be analogous.
   2. In addition, in the appendix, I didn’t see a curve in the ablation study with *both* smoothness and cycle-consistency loss removed.
   3. Not presenting this result with the other methods and putting it in the appendix gives an incomplete and possibly misleading impression to someone quickly eyeballing results like many readers tend to do. I think it is highly detrimental to the results presentation.
18. On the COIL-100 dataset, it is not clear whether during training you used interpolations only between objects of the same class or between classes as well.
19. “Results on other datasets can be seen in the Appendix.” What other datasets? I seem to only see results for COIL-100 and the pole dataset.
20. In addition it would be interesting to see what happens for interpolations between classes.

=====POST-REBUTTAL COMMENTS======== I thank the authors for the response and the efforts in the updated draft. Most of my queries were clarified and I raised my rating accordingly. I understand the authors view that some of my queries fall outside their desired scope for the paper, however I still think the paper could benefit from such contents.

---

> ### Author Response · Authors · 2020-11-23
> **Response to Reviewer 2 - Part I**
>
> **Q2.1**: No direct comparison to SOTA baselines, namely GAIA and AMR.
>
> **A2.1**: We have added a direct comparison to GAIA and AMR. See figures 5,6,7,8 in the paper and Figures 9,10,11 in the Appendix. The comparisons include visual inspections as well as quantitative comparisons. These comparisons establish the superiority of our proposed framework over previously suggested methods including GAIA and AMR.
>
> **Q2.2**: There’s no measure of the effects of the better conditioned latent space on downstream tasks such as a classification on SVHN or CIFAR10 like some other works in the domain checked.
>
> **A2.2**: This is a very good point as we believe the suggested scheme can assist not only in generating smooth and faithful interpolations but also to provide a powerful regularization term, that may enable generating new realistic examples for data augmentation, and it might be useful also for downstream tasks. However, we believe these verticals are outside the scope of this paper. In this paper, we present the general ideas and justify their contributions. We plan to evaluate the effects of the conditioned latent space on downstream tasks in future works.
>
> **Q2.3**: Novelty is not clearly identified: a good place would be the end of section 1.
>
> **A2.3**: We have added a description of our contributions and novelty at the end of section 1.
>
> **Q2.4**: Ablation should really be in the main section of the paper since it’s important. Figures (3) and (4) which didn’t really add to my comprehension could be removed to save space.
>
> **A2.4**: We felt that figure (3) offers invaluable intuition to the problem we aim to solve and to the methods we developed to address this problem. We believe Figure (3) makes the paper clearer. Figure (4) demonstrates the implementation of our network and we believe it is essential. Due to space limitation and prioritization, we decided not to include the ablation study in the main body of the paper, however, we added an additional reference to the ablation study as we believe it further manifests the justification for each loss component.
>
> **Q2.5**: “This regularization ... can also be used as a general regularization technique to avoid overfitting or to produce new samples for data augmentation.” It would be good to see it demonstrated in results.
>
> **A2.5**: As stated above, this is outside the scope of this paper and we plan to test the contribution of our technique as a general regularizer as well as an augmentation strategy in future works.
>
> **Q2.6**: In several places, you cite auto-encoders but really refer to variational auto-encoders, not crediting earlier works.
>
> **A2.6**: Corrected.
>
> **Q2.7**: “Researchers have demonstrated the ability to interpolate between data points by decoding a convex sum of latent vectors (Shu et al., 2018) ...” I believe there’s prior art before that.
>
> **A2.7**: Corrected.
>
> **Q2.8**: In several places, the word “reliable” is used, I suppose you meant “realistic”.
>
> **A2.8**: Corrected.
>
> **Q2.9**: “... while providing a convex latent manifold with a bijective mapping between the input and latent spaces.” It seems counter-intuitive to me, the input space is much larger than the latent space, how can it be bijective?
>
> **A2.9**: We meant between the input manifold and the latent manifold. We added this clarification in the paper. Note that the intrinsic dimensionality of the manifold is much lower than the dimensionality of the input space.
>
> **Q2.10**:  The cycle consistency loss tries to map $f(g(z_{i,j,α}))$ to $z_{i,j,α}$, in other words it looks surjective only. …If you are referring to the mapping from the latent space onto itself by $f(g(.))$, then the term identity mapping would seem clearer to me.
>
> **A2.10**: If the mapping f is surjective, then you can find $x_i$ and $x_j$, $(x_i \neq x_j )$ where $f(x_i)=f(x_j)=z$.  But $g(z)$ cannot map to $x_i$ and $x_j$ simultaneously. That is, to satisfy both the cycle-consistency and the adversarial constraints, the mapping from $x$ to $z$ must be bijective.
>
> **Q2.11**: “Credibility” I don’t understand why it is written in such a complicated manner.
>
> **A2.11**: Credibility formulation was simplified.
>
> **Q2.12**: Concerning the smoothness loss $L_s$, it seems to be derived from the k-Lipschitz constraint (3), But in its implementation, it’s really 0-Lipschitz, it would be great to clarify this aspect.

---

> > ### Author Response · Authors · 2020-11-23
> > **Response to Reviewer 2 - Part II**
> >
> > **A2.12**: If it were 0-Lipschitz then $x_i=x_{i,j}(0)=x_{i,j}(1)=x_i$ and this is impossible due to the reconstruction constraint  $x_{i,j}(1)=x_j$.  The $L_2$ minimization of the derivatives over $\alpha \in [0,1]$ tends to distribute the magnitude of the derivative uniformly between $\alpha \in [0,1]$.
> >
> > **Q2.13**: concerning $L_s$, the formulation seems odd. I’d expect it to be written
> > as $∑||\frac{d\hat{x}_{i,j}(α)}{d\alpha} (n/M)||^2$ since in its current form α does not appear in the numerator
> >
> > **A2.13**: We corrected the notation.
> >
> > **Q2.14**: If I’m correct you’re simply minimizing the pixel distance between successive images on the interpolation line.
> >
> > **A2.14**: Your interpretation is correct, but we apply the analytical derivative (by backpropagating through $\alpha$) and do not apply a numerical derivative as you indicated.
> >
> > **Q2.15**: In 2.2 JUSTIFICATION FOR THE PROPOSED APPROACH”. I found it was already justified by the intro to section 2 for the most part.
> >
> > **A2.15**: We find that the definition of admissible interpolation in part (2) is specified in (2.1) with regards to the implementation details of our architecture. In (2.2) we connect the theoretical part with the implementation details, and we believe it is crucial to the motivation behind our network design.
> >
> > **Q2.16**: On the topic of $L_c$, it is justified as “the cycle-consistency loss $L_c$ forces the encoder- decoder architecture to map linearly interpolated latent vectors onto the image manifold ...”. I don’t see why it should. … This could use clarification.
> >
> > **A2.16**: We fixed the cumbersome phrasing in the text.
> >
> > **Q2.17**: In the comparison to other methods, did you use the same number as training examples?
> >
> > **A2.17**: All tested architectures (AEAI, AAE, ACAI, BetaVAE, AMR, GAIA) were trained using the same number of samples, roughly the same number of epochs, and using an 80-20 split (training, validation). We have included clarifications regarding this issue in the paper (4.1).
> >
> > **Q2.18**: “The two images $(xi, xj)$ are encoded by the shared-weight encoder...”. I didn’t follow what the weights are shared with.
> >
> > **A2.18**: The architecture uses a single encoder and a single decoder. We have cleared it in the paper.
> >
> > **Q2.19**: The loss terms are called $L_{a,c,s}$ but the weighting hyper-parameters are called $\lambda_{1,2,3}$. Using the same letter as their corresponding loss term instead of 1,2,3 would make reading more friendly.
> >
> > **A2.19**: Lambda subscripted fixed to match the loss subscript.
> >
> > **Q2.20**: “The GAIA method of Sainburg et al. (2018) is similar in spirit to the AMR framework.” The GAIA method came first, so this is the other way around (B is similar in spirit to A, if B came after A).
> >
> > **A2.20**: GAIA is now presented before AMR.
> >
> > **Q2.21**: “but also ensure a diverse generation ... while avoiding mode collapse “. How does it ensure it? could you give more insights on what causes it?
> >
> > **A2.21**: Due to the bijective mapping we avoid mode-collapse (surjective) when interpolating along $\alpha=[0,1]$. This causes the interpolation to generate diverse samples.
> >
> > **Q2.22**: The results section comes just after Related Work and it’s striking that there’s no comparison to the two methods yours is most similar to and that were covered sentences.
> >
> > **A2.22**: We have added a direct comparison to GAIA and AMR.
> >
> > **Q2.23**: I understand you claim that “Comparisons with AMR and GAIA methods are analogous to the ablation study presented in the Appendix, but it’s not clear whether you ran the experiment and observed it was indeed analogous or whether it should be analogous.
> >
> > **A2.23**: Same as A2.22.
> >
> > **Q2.24**: Not presenting this (ablation) result with the other methods and putting it in the appendix gives an incomplete and possibly misleading impression … I think it is highly detrimental to the results presentation.
> >
> > **A2.24**: This is a question of prioritization. We feel that our comparison with the competitive approaches better justifies and delivers the suggested approach and should remain in the main paper. There is no room left for presenting both, the ablation and the comparison results. We added references in the paper for the ablation study in the Appendix.
> >
> > **Q2.25**: On the COIL-100 dataset, it is not clear whether during training you used interpolations only between objects of the same class or between classes as well.
> >
> > **Q2.26**: it would be interesting to see what happens for interpolations between classes.
> >
> > **A2.25+2.26**: We perform inter-class interpolations only. We made it clear in the paper. This is along the mainline of the paper that we interpolate along with manifold data.  Interpolating along multi-modal distributions might give good results, but it is outside of the scope of this paper.
> >
> > **Q2.27**: Results on other datasets can be seen in the Appendix.” What other datasets? I seem to only see results for COIL-100 and the pole dataset.
> >
> > **A2.27**: This was a typo. Corrected.

---

### Official Review · AnonReviewer1 · 2020-10-29
**Interpolating Autoencoder**

**Rating:** 5
**Confidence:** 4

**Review:**

This paper focused on developing a new regularization technique for autoencoders, which shapes the latent representation to follow a manifold that is consistent with the training images and that drives the manifold to be smooth and locally convex. The authors suggest that the manifold structure of continuous data must be considered to include the geometry and shape of the manifold. The new interpolation regularization mechanism consists of an adversarial loss, a cycle-consistency loss, and a smoothness loss. So the architecture of the proposed model includes a standard autoencoder, a discriminator and the loss mentioned above.

Strengths: The motivation of using four losses were clearly described and the architecture of the model was straightforward. The authors tested the proposed AEAI method on one synthetic plot dataset and COIL-100 dataset and compared the results with three other models. The effectiveness of the proposed technique is tested by both visual inspection and comparing the reconstruction error against the available ground-truth images. The authors also examined the transition smoothness from one sample to the other on both datasets. The comparison of the alpha values showed that the proposed technique did generate more smooth results than other methods. By visual inspection, the results looked pretty good for the proposed technique while other interpolations methods change abruptly between modes and introduce smallartifacts during reconstruction. The evaluation of MSE also showed the proposed technique achieved lower MSE. In general, I think the analysis of the proposed technique was convincing, and the authors also provided ablation study with smoothness and cycle-consistency losses missing, showing the necessity of adding these losses.

Weakness: The proposed model is a mixture of known techniques, without a unifying theme. The authors can try to compare the latent space directly with manifold learning techniques or autoencoders that directly penalize using manifold learning. One example is the recent paper at ICML https://arxiv.org/abs/2002.04881 (Chen et al. "Learning Flat Latent Manifolds with VAEs." ICML 2020) or https://www.sciencedirect.com/science/article/abs/pii/S1063520317300957 (Mishne et al, Diffusion Nets, Applied and Computational Harmonic Analysis.") The authors also don't quantify the manifold learning capability of the latent space. A potential metric for this is the Denoised Manifold Affinity-Preservation Metric (DeMAP) proposed in Moon et al., Nature Biotechnology 2019. I would also suggest visualizations of the latent space using the PHATE technique from the same paper.

I also feel that the authors should try a more irregular/noisy dataset to further show the effectiveness of the model.

---

> ### Author Response · Authors · 2020-11-24
> **Response to Reviewer 1**
>
> **Q1.1**: The proposed model is a mixture of known techniques without a unifying theme.
>
> **A1.1**: We disagree with this comment and believe that our paper proposes a novel scheme whose strength and innovation stems from the right mixture of the necessary components, although each component might be a known technique. The fact that such a complete method has not been suggested previously and the superiority of our results compared to previously proposed methods indicate that the suggested framework is valid and novel. With respect to a “unifying theme”: At the beginning of Sec 2 we present a complete and coherent requirement for an admissible interpolation, and in Sec 2.1 and 2.2 we provide a unified and justified scheme where the contribution of each component is well justified.
>
> **Q1.2**: The authors can try to compare the latent space directly with manifold learning techniques or autoencoders that directly penalize using manifold learning
>
> **A1.2**: Manifold learning techniques aim at locality preserving embedding, where affinity matrix $K$ is defined between input points $x_i$ and $x_j$ using a pre-defined kernel $k(x_i,x_j)$. A distance is defined between each pair of points, $d(x_i,x_j)$, using diffusion distance, shortest path in the K-NN graph, or any other definition assisting the affinity matrix K. The Manifold learning-based autoencoder aims at optimizing the latent space in the sense that distances in the input space should be proportional to distances in the latent space. I.e. $d(x_i,x_j) \approx |z_i-z_j|^2$. Although the latent interpolation scheme can assist autoencoders based manifold learning, we do not see a strong connection between the two schemes due to the following reasons:
>
> 1. Our ultimate goal is a realistic interpolation, while manifold learning does not take this objective as an explicit goal.
> 2. Following (a), the adversarial loss, cycle consistency, and smoothness are not explicit objectives of a manifold learning encoder mapping.
> 3. Manifold learning aims at embedding that will satisfy a pre-defined distance $d(x_i,x_j)$. Our scheme does not use any pre-defined affinity kernel and the encoder applies mapping in a completely unsupervised manner.

---

### Author Response · Authors · 2020-11-23
**Note to Reviewers: Thank You for the Detailed Review Process**

We would like to thank the reviewers for their valuable comments and useful suggestions. Based on the reviewer comments we made several changes to the paper. We believe the new changes improved the paper and made its contribution clearer.

In particular, we have added a direct comparison to the competitive methods of GAIA and AMR. The comparisons include visual inspections as well as quantitative comparisons. These comparisons established the superiority of our proposed framework over previously suggested methods including GAIA and AMR.

---

### Decision · Program_Chairs · 2021-01-07
**Final Decision**

**Decision:**

Reject

**Comment:**

The authors propose a technique called Autoencoder Adversarial Interpolation (AEAI). The key idea is to train autoencoder architectures that explicitly "shapes" trajectories in the encoder (latent) space to correspond to smooth geodesics between data points. This is achieved by a combination of several loss terms that are fairly intuitive. The authors empirically justify each term via ablation studies on simple datasets.

Initial review scores had wide variance. The reviewers liked the overall approach as well as the clarity with which the theory and experiments were presented, but raised several concerns. The authors provided succinct responses that seem to have satisfied the reviewers on average.

Unfortunately, after having carefully read this paper (and the authors' responses), I have to go against the wishes of the majority of the reviewers, and recommend a reject. My two main concerns are as follows:
a) The synthetic pole dataset, as well as the COIL-100 dataset, are far too simplistic to evaluate performance. It is now standard to report results on considerably more challenging datasets.
b) Echoing R2 -- the authors should articulate why a shaped latent space should actually matter in applications, beyond giving intuitive(I guess?) visualizations and reconstruction error curves. Results on downstream tasks may be one avenue to achieve this.